# Clinical Features of Diabetes Mellitus on Rheumatoid Arthritis: Data from the Cardiovascular Obesity and Rheumatic DISease (CORDIS) Study Group

**DOI:** 10.3390/jcm12062148

**Published:** 2023-03-09

**Authors:** Fabio Cacciapaglia, Francesca Romana Spinelli, Elena Bartoloni, Serena Bugatti, Gian Luca Erre, Marco Fornaro, Andreina Manfredi, Matteo Piga, Garifallia Sakellariou, Ombretta Viapiana, Fabiola Atzeni, Elisa Gremese

**Affiliations:** 1Department of Precision and Regenerative Medicine and Jonian Area, Università Degli Studi di Bari Facoltà di Medicina e Chirurgia, 70124 Bari, Italy; 2Dipartimento di Scienze Cliniche Internistiche, Anestesiologiche e Cardiovascolari—Reumatologia, Università Degli Studi di Roma La Sapienza, 00185 Roma, Italy; 3Rheumatology Unit, Department of Medicine and Surgery, University of Perugia, 06100 Perugia, Italy; 4Department of Internal Medicine and Therapeutics, University of Pavia, 27100 Pavia, Italy; 5Division of Rheumatology, Fondazione IRCCS Policlinico San Matteo, 27100 Pavia, Italy; 6Dipartimento di Medicina, Chirurgia e Farmacia, Università Degli Studi di Sassari, 07100 Sassari, Italy; 7Rheumatology Unit, Azienda Ospedaliera Universitaria Policlinico of Modena, 41121 Modena, Italy; 8Rheumatology Unit, Department of Medical Sciences and Public Health, University of Cagliari, University Clinic AOU, 09042 Cagliari, Italy; 9Istituti Clinici Scientifici Maugeri, 27100 Pavia, Italy; 10Rheumatology Unit, Department of Medicine, University and Azienda Ospedaliera Universitaria Integrata of Verona, 37126 Verona, Italy; 11Rheumatology Unit, Department of Experimental and Internal Medicine, University of Messina, 98122 Messina, Italy; 12Division of Clinical Immunology, Fondazione Policlinico Universitario A. Gemelli—IRCCS, Università Cattolica del Sacro Cuore, 00168 Rome, Italy

**Keywords:** diabetes mellitus, rheumatoid arthritis, biological drugs

## Abstract

Rheumatoid arthritis (RA) and diabetes mellitus (DM) are linked by underlying inflammation influencing their development and progression. Nevertheless, the profile of diabetic RA patients and the impact of DM on RA need to be elucidated. This cross-sectional study includes 1523 patients with RA and no episodes of cardiovascular events, followed up in 10 Italian University Rheumatologic Centers between 1 January and 31 December 2019 belonging to the “Cardiovascular Obesity and Rheumatic DISease (CORDIS)” Study Group of the Italian Society of Rheumatology. The demographic and clinical features of DM RA patients were compared to non-diabetic ones evaluating factors associated with increased risk of DM. Overall, 9.3% of the RA patients had DM, and DM type 2 was more common (90.2%). DM patients were significantly older (*p* < 0.001), more frequently male (*p* = 0.017), with a significantly higher BMI and mean weight (*p* < 0.001) compared to non-diabetic patients. DM patients were less likely to be on glucocorticoids (*p* < 0.001), with a trend towards a more frequent use of b/ts DMARDs (*p* = 0.08), and demonstrated higher HAQ (*p* = 0.001). In around 42% of patients (*n* = 114), DM diagnosis preceded that of RA. Treatment lines were identical in diabetic and non-diabetic RA patients. DM is a comorbidity that may influence RA management and outcome. The association between DM and RA supports the theory of systemic inflammation as a condition underlying the development of both diseases. DM may not have a substantial impact on bDMARDs resistance, although further investigation is required to clarify the implications of biological therapy resistance in RA patients.

## 1. Introduction

Rheumatoid arthritis (RA) is a chronic autoimmune disease characterized by both local and systemic inflammation, ultimately leading to joint damage [1]. The global prevalence of RA has been estimated at around 0.46% from studies conducted between 1980 and 2019 [2], and a similar prevalence (0.48%) is reported for the Italian population [3].

The prevalence of RA is highest around the fifth decade of life and is generally higher in females than in males. [1,4]. Several inflammatory pathways are involved in the pathogenesis and development of RA, including pro-inflammatory cytokines such as tumor necrosis factor α (TNFα), interleukin 1 (IL-1), IL-6, and immune cells such as autoreactive CD4+ T cells, B cells and macrophages [1,5]. Local synovial inflammation characterizes the early stages of RA, subsequently expanding and becoming systemic due to the diffusion of inflammatory cytokines and immune cells. Systemic inflammation is supposed to damage tissues other than joints, such as the cardiovascular (CV) system [6]. Indeed, a high risk of CV events was observed in this patient population [7,8], and CV disease is the main contributor to the excess mortality that characterizes people with RA. [1].

Diabetes mellitus (DM) is a metabolic disease that affects over 530 million people globally, a number that is expected to rise in the next 25 years [9]. An estimated 3 million Italians (5% of the population and up to 16% of those over 65) have diabetes [10]. In comparison to non-diabetic persons, diabetic patients, especially females and those with type 2 DM, have a two- to four-fold greater risk for CV events [11,12]. Diabetes patients frequently have obesity, insulin resistance, and uncontrolled hyperglycemia, all of which increase the risk of CV disease. Both type 1 and type 2 DM have inflammation-related pathogenesis, and some of the mediators of inflammation are also present in RA, suggesting a potential connection between these two diseases [13]. About 20% of RA patients have DM as a comorbidity, and RA is linked to a higher risk of DM development [14,15].

Previous longitudinal studies have demonstrated a significantly higher and at least double CV risk in RA patients compared to the general population, comparable to the magnitude of CV events observed in type 2 DM [16,17]. Furthermore, the CARRÉ (CARdiovascular research and RhEumatoid arthritis) study found, in a prospective cohort examining CVD in long-term RA, an even higher risk in patients with RA compared to DM, and the association of RA with the two conditions of DM or insulin resistance was associated with the highest risk of developing CVD [18]. Therefore, treating both conditions with appropriate therapy is essential to slow the progression of these diseases and avoid CV events.

The profile of diabetic RA patients and the impact of DM on RA progression are still debated. Therefore, the present study aimed to analyze the demographic and clinical features of RA patients with and without diabetes, describing possible differences in therapeutic management eventually responsible for a different profile of CV complications.

## 2. Patients and Methods

RA patients included in this cross-sectional study fulfilled the 2010 American College of Rheumatology (ACR)/EULAR classification criteria [19] and were regularly evaluated and followed up with in 10 Italian University Rheumatologic Centers. The database of the “Cardiovascular Obesity and Rheumatic DISease (CORDIS)” Study Group was screened for consecutive patients with a medical examination between 1 January and 31 December 2019 [7]. The CORDIS study group is a non-profit study group established within the Italian Society of Rheumatology on the initiative of academic rheumatologists interested in the study of CV risk in rheumatic diseases. This group aims to understand the interrelation between inflammation, autoimmunity and CVD by collecting epidemiological, clinical and laboratory data from Italian patients with musculoskeletal rheumatic diseases. Patients were excluded if they presented prior CV events (acute coronary syndrome, that is, ST- and non-ST elevation myocardial infarction, coronary revascularization and unstable angina), stable angina pectoris, ischemic stroke and peripheral artery disease (with or without revascularization procedures), retrieved by review of medical charts [7].

Patients enrolled were characterized by the following clinical and serological data: age, sex, smoking status, body mass index (BMI), systolic and diastolic blood pressure, lipid profile, presence of DM, and hypertension. Dyslipidemia was defined by the use of lipid-lowering drugs and/or low-density lipoprotein (LDL) cholesterol targets, based on each patient’s CV risk, as recommended by the ESC/EAS guidelines for the management of dyslipidemia. [20]. Patients with hypertension had either a history of hypertension or made use of blood-pressure-lowering drugs. DM was defined based on previous medical history and/or the use of oral or parenteral hypoglycemic medications or insulin as reported [21].

Information on current treatments—ongoing anti-hypertensive, lipid-lowering therapies, and anti-rheumatic drugs, including conventional synthetic (cs) disease-modifying anti-rheumatic drugs (DMARDs), biologic (b) and targeted synthetic (ts) DMARDs and glucocorticoids (mean weekly dose since diagnosis and the current daily dose of prednisone or equivalent)—were recorded. In addition, rheumatoid factor (RF) and anti-citrullinated peptide antibodies (ACPA) were evaluated via serological assays [7].

At baseline, disease-specific factors such as disease duration, C reactive protein (CRP), disease activity index based on 28-joint evaluation (DAS28), and Clinical Disease Activity Index (CDAI) as measures of disease activity and the Health Assessment Questionnaire (HAQ) disability index as the function index were evaluated [22].

The present study was conducted according to the ethical guidelines of the Declaration of Helsinki and was approved by the local Ethical Committee as part of the GISEA Registry protocol (approval number DG-624/2012). At the start of the observational period, written informed consent was obtained from all patients.

### Statistical Analysis

Data were expressed as mean ± standard deviation (SD) or 95% confidence intervals (95% CI) when appropriate. Differences in continuous variables were evaluated using the paired *t*-test and/or repeated analysis of variance (ANOVA) followed by the Bonferroni post hoc test. Fisher’s exact test was used for categorical data to assess differences between the two groups.

A *p*-value < 0.05 was considered statistically significant, while variables with a *p*-value < 0.1 in descriptive statistical analysis were evaluated with a logistic regression model and univariate analysis. Subsequently, variables that achieved statistical significance (*p* < 0.05) in univariate analysis were then included in a multivariate logistic regression model adjusted for age, sex, and disease duration.

All analyses were performed using the SPSS statistic program (version 21—IBM software, New York, NY, USA).

## 3. Results

### 3.1. Patient Demographics

Overall, 1523 patients diagnosed with RA and with no previous CV events were included in the analysis.

Demographics and clinical features of patients are reported in Table 1. Diabetic patients were 9.3% of the total population, were significantly older than non-diabetic patients (*p* < 0.001), and were more frequently male (*p* = 0.017). In addition, diabetic patients had a significantly higher BMI and mean weight (*p* < 0.001) compared to non-diabetic patients. The prevalence of obesity, hypertension and use of antiplatelet drugs was also significantly higher in diabetic patients than in their non-diabetic counterparts (*p* < 0.001). Of note, 613 out of 649 (94.4%) of our RA patients were on anti-hypertensive treatment with no significant differences among diabetic and non-diabetic patients (88 out of 93 vs. 525 out of 556—*p* = 0.93). Considering the disease-specific factors, diabetic patients demonstrated higher HAQ (*p* = 0.001) despite a similar mean duration of the disease. Patients with DM were less frequently on glucocorticoids (*p* < 0.001) and demonstrated a trend towards a more frequent use of b/tsDMARDs (*p* = 0.08).

After logistic regression analysis adjusted for age, sex, and disease duration, the variables BMI, weight, obesity, hypertension, anti-platelets drugs, HAQ, and bDMARDs use were independently associated with an increased risk of DM in RA patients (Table 2).

### 3.2. Rheumatoid Arthritis Patients with Concomitant DM

Specific clinical features of diabetic patients are reported in Table 3.

The majority of RA diabetic patients (90.2%) suffered from type 2 DM. Data about the time of onset, available for 114 patients, indicated that in around 42% of patients, DM diagnosis preceded the diagnosis of RA. Diabetic-related complications were present in 29/121 patients, and oral hypoglycemic drugs were the most used for the treatment of DM.

## 4. Discussion

This cross-sectional study on a large cohort of RA patients included in the database of the “Cardiovascular Obesity and Rheumatic DISease (CORDIS)” Study Group of the Italian Society of Rheumatology indicated a prevalence of DM around 10% among RA patients, suggesting that DM, as a frequent comorbidity, may influence the management and outcome of RA. When compared to non-diabetic RA patients, diabetic RA patients were older, more usually men, had a greater incidence of being overweight and obese, and were more frequently hypertensive. All these characteristics were confirmed after adjustment for age, sex and disease duration. Notably, diabetic RA patients were less likely to be on corticosteroids. However, after adjusting for sex, age and disease duration, we found that diabetic RA patients had a significantly higher probability of being on biological drugs, as evidenced by a significantly lower HAQ-DI among non-diabetic RA patients compared to diabetic ones, who are known to suffer from more comorbidities. This observation could be linked to a more severe disease course in RA patients with DM.

The overall prevalence of diabetes mellitus (DM) was higher in our cohort of RA patients compared to osteoarthritis control subjects [7] and the general Italian population [10], supporting the theory that systemic inflammation and the concurrent use of glucocorticoids may contribute to the development of both diseases. There is a great deal of geographical heterogeneity in the data that are currently available [23,24] regarding the prevalence of DM in the RA community, ranging from 2% in Ireland to 20% in a German study [14].

A recent meta-analysis found that RA patients have a 23% greater chance of developing DM [15]. Such inconsistency may be explained by differences in environmental exposure, lifestyle-related factors, geographic setting, study design, DM definition criteria (ICD-10 code, prescription of hypoglycemic drugs or insulin, physician diagnosis, fasting glucose concentration), selection of the comparison group, and characteristics of populations studied. However, studies show a consistent, significantly higher incidence of DM in RA patients compared to the general population [25,26,27]. Furthermore, when compared to the general population, type 2 DM was more prevalent in our RA patient cohort, which was previously reported by an Italian study [28].

Interestingly, we observed that DM onset, for about 58% of patients, occurred after the diagnosis of RA. The onset of DM after RA may depend on different ages but can also suggest a role for glucocorticoids on the glycol-metabolic functions of such patients, as previously described [29]. In RA patients, glucocorticoid therapy has been associated with an increase in insulin resistance, a decrease in insulin sensitivity, and a higher risk of type 2 DM [30]. However, in our study, diabetic patients were less likely to be taking glucocorticoids at enrollment compared to non-diabetic RA patients, which may indicate an attempt by doctors to improve glycemic management [31]. This hypothesis is supported by the significantly higher prevalence of biological drugs prescribed to our diabetic patients as steroid-sparing medications, which is consistent with the EULAR recommendation to taper glucocorticoid doses in RA patients “as rapidly as clinically feasible” to prevent long-term side effects such as hyperglycemia and DM [32]. Additionally, EULAR guidelines for managing CV risk in people with inflammatory arthritis include restricting the use of glucocorticoids, which are linked to an increased risk of CV events and all-cause mortality, particularly in those with concurrent DM [23,33,34]. We cannot rule out the possibility that the cumulative dose of glucocorticoids in our population may have induced DM; at the time of data collection, the dose of prednisone prescribed was significantly lower in patients with RA and DM than in non-diabetic patients, possibly as a result of this comorbidity. Moreover, one should consider the beneficial effects of some conventional drugs, such as methotrexate and hydroxychloroquine, on glucose metabolism as well as the various effects of bDMARDs, such as tumor necrosis factor, IL-6, or IL-1 antagonists, on insulin resistance [30].

The increased risk of developing DM may be due to an inflammatory condition associated with RA, with the over-activation of TNFα and IL-6 pathways contributing to an increase in insulin resistance and causing DM [35]. Intriguingly, independent of BMI or specific RA therapy, such as the use of glucocorticoids, higher baseline disease activity and elevated levels of pro-inflammatory cytokines, including IL-1 and IL-6, are associated with a significantly higher risk of incident DM in RA patients [36]. This confirms the significant relation between systemic inflammation, insulin resistance, and the risk of developing DM in RA patients and highlights the significance of achieving optimal disease activity control in these patients.

The results of the current study are noteworthy in that, although diabetic patients with RA received various pharmaceutical treatments, DM was not adequately controlled in about one-third of them, as evidenced by the average HbA1c values. As evidenced by a recent retrospective study reporting that around one-third of RA patients had baseline HbA1c levels ≥7, DM is underdiagnosed and not appropriately managed in diabetic RA patients, just as blood pressure is in RA patients with hypertension [37]. This suggests that it is crucial for RA patients to be aware of their glycometabolic profile, which needs to be managed appropriately. Under the current guidelines, metformin should be used as a first-line treatment for DM in RA patients or sodium-glucose cotransporter 2 inhibitors (such as empagliflozin) or glucagon-like peptide 1 receptor in those who have established CV disease or are at greater risk for developing it in the future [38].

The low DAS28 and CDAI scores within our group of patients demonstrate that several therapies recommended by various centers were effective in controlling RA activity. It is interesting to note that the number of treatment lines was identical in diabetic and non-diabetic RA patients, indicating that DM may not have a substantial impact on bDMARDs resistance. However, more investigation is required to clarify the implications of biological therapy resistance in RA patients [39,40].

Our analysis supports previous cohort study findings [41] that comorbidities are more common in diabetic patients, and it shows that DM risk in these individuals should not be attributed solely to inflammation. Particularly, our diabetic RA patients had significantly higher rates of hypertension and overweight/obesity. Such comorbidities are closely linked to DM [42,43]. Data on prevalence and control of hypertension in RA are discordant, showing higher prevalence but also underdiagnosis and higher or lower likelihood of antihypertensive treatment [38]. In our cohort, 649 (42.6%) of patients had a previous diagnosis of hypertension and 613 of them (94.4%) were taking anti-hypertensive drugs, with no differences among diabetic and non-diabetic patients. The coexistence of hypertension and diabetes is not fully unexpected; indeed, in the general population, hypertension is commonly associated with diabetes [44]. Moreover, RA patients also took glucocorticoids significantly more frequently, above all if non-diabetic. As is known, chronic glucocorticoid use may increase the risk of developing both hypertension and diabetes [45], and some anti-hypertensive agents can also improve glucose metabolism [46,47]. Comorbidities may have an impact on how patients with RA are managed and eventually their quality of life, which emphasizes the value of a customized therapeutic approach [48,49].

It is important to note some of our study’s shortcomings. To begin with, this is a cross-sectional study of a small sample; in addition, because this is an observational study, there are some missing data regarding DM characteristics and therapeutic care. Furthermore, this sort of study does not allow for speculation on the impact of treatment duration and the effect of RA treatments on DM outcome. The exclusion of patients with previous CV events might appear to be a bias. On the contrary, as the impact of a previous CV event on subsequent CV risk is well known, we opted for the evaluation of patients without a history of CV events to reduce the potential bias on the clinical management of RA and DM: patients with previous CV events have a peculiar therapeutic strategy and specific follow-up that might influence the clinical management of the underlying disease, both for RA and DM. The inclusion of consecutive RA patients treated according to local (Italian) indications in tertiary-level rheumatology centers may have selected a population with more severe disease. However, the fact that about 40% of the entire study population was treated with bDMARDs indicates there was a balanced proportion of patients that would certainly have been higher if only patients with a more severe disease had been selected. Finally, residual confounding factors such as diet adherence and physical activity were not considered in data interpretation.

Despite these drawbacks, the present study’s findings portray a real-life population indicative of all diverse Italian settings, allowing a more in-depth delineation of the characteristics and clinical management of DM as a comorbidity in RA patients. Our analysis’s findings undoubtedly show that better glucose metabolic profile control, concurrent comorbidities including obesity and hypertension, and improved awareness of the need for good DM management are needed to lower the risk of CV events in this population. More research is needed to understand the intricate link between inflammation, insulin resistance, glucose metabolic pathways, and the influence of concomitant therapies on DM.

## Figures and Tables

**Table 1 jcm-12-02148-t001:** Demographic and clinical features of RA patients.

	Overall	Diabetic	Non-Diabetic	*p*-Value
Patients, *n* (%)	1523	142 (9.3)	1381 (90.7)	
Age, years—mean (SD)	59.3 (11.7)	63.7 (10.9)	59.3 (11.7)	<0.001
Female, *n* (%)	1202 (78.9)	101 (71.1)	1101 (79.7)	0.017
Disease duration, months—mean (95% CI)	133.6 (128.2–139)	130.6 (112.2–149)	133.9 (128.3–139.6)	0.24
RF positivity, *n* (%)	953 (62.6)	82 (57.7)	871 (63.1)	0.21
ACPA positivity, *n* (%)	923 (60.6)	80 (56.3)	843 (61)	0.27
CRP mg/L, mean (95%CI)	6.9 (5.5–8.4)	7.4 (4.6–10.1)	6.9 (5.5–8.4)	0.11
HAQ, mean (95%CI)	0.78 (0.74–0.82)	1.01 (0.86–1.16)	0.76 (0.71–0.8)	0.001
DAS28-ESR, mean (95%CI)	3.2 (3.1–3.2)	3.3 (3.1–3.5)	3.2 (3.1–3.2)	0.28
CDAI, mean (95%CI)	8.8 (8.3–9.3)	9.8 (8–11.6)	8.8 (8.3–9.3)	0.40
DAS28-ERS < 2.6, *n*. (%)	493 (32.4)	47 (33.1)	446 (32.3)	0.85
CDAI ≤ 10, *n* (%)	1023 (67.2)	88 (62)	935 (67.7)	0.17
BMI, kg/m^2^—mean (SD)	25.7 (4.8)	27.7 (4.7)	25.5 (4.8)	<0.001
BMI > 30 kg/m^2^, *n* (%)	232 (15.2)	42 (29.6)	190 (13.7)	<0.001
Weight, kg—mean (SD)	68.2 (14.1)	73.1 (13.5)	67.7 (14)	<0.001
Height, cm—mean (SD)	163 (8)	163 (8)	160 (7)	0.16
Dyslipidemia, *n* (%)	890 (58.4)	79 (55.6)	811 (58.7)	0.48
Hypertension, *n* (%)	649 (42.6)	93 (65.5)	556 (40.3)	<0.001
Smoking, *n* (%)	293 (19.2)	25 (17.6)	268 (19.4)	0.60
Antiplatelet drugs, *n* (%)	156 (10.2)	34 (23.9)	122 (8.8)	<0.001
Anticoagulant agents, *n* (%)	34 (2.2)	5 (3.5)	29 (2.1)	0.27
Corticosteroids, *n* (%)	639 (42)	52 (36.6)	587 (42.5)	<0.001
Prednisone eq. dose (mg/day), mean (95%CI)	3.2 (2.9–3.5)	3.2 (2.5–3.9)	3.2 (2.9–3.5)	0.89
csDMARDs, *n* (%)	1034 (67.9)	100 (70.4)	934 (67.6)	0.76
b/tsDMARDs, *n* (%)	636 (41.8)	69 (48.6)	567 (41)	0.08
1st-line	297 (46.7)	33 (47.8)	264 (46.6)	0.90
2nd-line	146 (23)	12 (17.4)	134 (23.6)	0.40
≥3rd-line	193 (30.3)	24 (34.8)	169 (29.8)	0.24

**Table 2 jcm-12-02148-t002:** Logistic regression model.

	Univariate			Multivariate *		
	OR	95% CI	*p*-Value	OR	95% CI	*p*-Value
Age	1.03	1.02–1.05	<0.001	1.03	1.02–1.05	<0.001
Sex (female vs. male)	0.62	0.42–0.92	0.018	0.69	0.46–1.03	0.07
Disease duration	0.99	0.99–1.00	0.52			
BMI, kg/m^2^	1.09	1.05–1.13	<0.001	1.09	1.05–1.13	<0.001
BMI > 30 kg/m^2^	2.75	1.84–4.09	<0.001	2.82	1.87–4.24	<0.001
Weight, kg	1.03	1.01–1.04	<0.001	1.03	1.01–1.04	<0.001
Hypertension	2.54	1.74–3.71	<0.001	2.28	1.52–3.39	<0.001
Antiplatelets drugs	3.67	2.39–5.62	<0.001	3.04	1.95–4.72	<0.001
HAQ-DI	1.52	1.21–1.92	<0.001	1.48	1.16–1.90	0.002
Glucocorticoids (yes/no)	0.78	0.56–1.12	0.177			
b/tsDMARD vs. csDMARD	1.36	0.96–1.9	0.08	1.6	1.11–2.30	0.012

* Adjusted for age, sex and disease duration. Dependent variable: DM.

**Table 3 jcm-12-02148-t003:** Clinical features of 142 RA patients with DM.

Feature	Value
Type 1/2 DM, *n* (%)	14/128 (9.8/90.2)
Patients with DM onset before RA *, *n* (%)	48 (42.1)
A1c hemoglobin **	
Percentage, mean (95%CI)	7.2 (6.9–7.4)
mmol/mol, mean (95%CI)	54.9 (52.4–57.5)
A1c hemoglobin not on target mmol/mol, *n* (%)	26 (32%)
DM complications, *n* ^§^ (%)	29 (24)
Nephropathy	9 (7.4)
Retinopathy	9 (7.4)
Peripheral arterial obliteration/diabetic foot	8 (6.6)
Neuropathy	2 (1.6)
Type of treatment, *n* ^ꝉ^ (%)	
Insulin	31 (23.8)
Oral hypoglycemic drugs	103 (79.2)
Biguanides	76 (73.8)
GLP-1 receptor agonist	7 (6.9)
Sulfonyl ureas	4 (3.9)
DPP-4 inhibitors	3 (2.9)
SGLT2 inhibitors	3 (2.9)
Alpha-glucosidase inhibitors	2 (1.9)

* Data available for 114 patients; ** data available for 81 patients; ^§^ data available for 121 patients; ^ꝉ^ data available for 130 patients.

## Data Availability

The data presented in this study are available on request from the corresponding author.

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
