# Peer review of "Clinical Features of Diabetes Mellitus on Rheumatoid Arthritis: Data from the Cardiovascular Obesity and Rheumatic DISease (CORDIS) Study Group"

_jcm, 2023, doi:10.3390/jcm12062148_

Round 1
Reviewer 1 Report
Overall a well conducted and reported study on moderately interesting topic.
Some comments:
1. Lines 105, 106: "DM was defined based on previous medical history and/or the use of oral hypoglycemic medications or insulin as reported [18]. "
GLP1 agonists are not oral (used by7% of DM patients in the study)
2. There was a higher prevalence of hypertension among patients with DM. Could it be due to increased screening for hypertension in this group of patients? It is known form previous studies (eg Panoulas 2008) that patients with RA have increased blood pressure but less confirmed hypertension/ lower proportion of blood pressure medication usage. (in your study having hypertension was defined by either a history of hypertension or use of blood pressure-lowering drugs)
3. Interestingly patients with DM had higher disease activity and lower quality of life but at the same time lower proportion of having poor prognostic factors- antiCCp/RF positivity, no significant difference in mean CRP level between the goups, how do you expain that? Could it be due to the association of higher BMI and more pain? Only composite disease activity measures are reported- do you have data on the number of painful and swollen joints? This should be discussed in more detail.
Author Response
Dear Reviewer,
We thank you for your suggestions, you can find the answer to your comments.
Best Regards,
Fabio Cacciapaglia, MD PhD
REVIEWER #1
Overall a well conducted and reported study on moderately interesting topic.
Some comments:
- Lines 105, 106: "DM was defined based on previous medical history and/or the use of oral hypoglycemic medications or insulin as reported [18]. "
GLP1 agonists are not oral (used by7% of DM patients in the study)
Answer:
According to your suggestion, we modified as follows: “… oral or parenteral hypoglycemic medications …””
- There was a higher prevalence of hypertension among patients with DM. Could it be due to increased screening for hypertension in this group of patients? It is known form previous studies (eg Panoulas 2008) that patients with RA have increased blood pressure but less confirmed hypertension/ lower proportion of blood pressure medication usage. (in your study having hypertension was defined by either a history of hypertension or use of blood pressure-lowering drugs)
Answer:
Data on prevalence and control of hypertension in RA are discordant, showing higher prevalence but also underdiagnosis and higher or lower likelihood of antihypertensive treatment (Semb AG Nat rev Rheumatol 2020). In our cohort 649 (42.6%) of patients had a previous diagnosis of hypertension and 613 of them (94.4%) were taking anti-hypertensive drugs, with no differences among diabetic and non-diabetic patients. The coexistence of hypertension and diabetes is not fully unexpected; indeed, in the general population, hypertension is commonly associated with diabetes (de Boer IH, Diabetes Care 2017). Moreover, RA patients also took glucocorticoids significantly more frequently, above all if non-diabetic. As is known, chronic glucocorticoid use may increase the risk of developing both hypertension and diabetes (Bergstra SA, ARD 2023), and some anti-hypertensive agents can also improve glucose metabolism (Suksomboon N, J Clin Pharm Ther 2012; Yao J PLos One 2021). Thanks to the reviewers suggestion, we comment more on this point in the Discussion section.
- Interestingly patients with DM had higher disease activity and lower quality of life but at the same time lower proportion of having poor prognostic factors- antiCCp/RF positivity, no significant difference in mean CRP level between the goups, how do you expain that? Could it be due to the association of higher BMI and more pain? Only composite disease activity measures are reported- do you have data on the number of painful and swollen joints? This should be discussed in more detail.
Answer:
The point you raised is pertrinent. However, as reported in Table 1 only the functional status measured by the HAQ disability index was significanlty different. If higher BMI is clearly associate with the disease actitivy, the relationship with the disability is less clear. We could speculate on an association between BMI and pain accounting for worse HAQ scores in diabetic overweight/obese patients, but it would be only speculation.
Reviewer 2 Report
Interesting work on a large number of RA-patients from Italian academic centres. Interesting data on higher BMI and lower GC use in RA-patients with DM, as expected.
major comments:
in the introduction it is stated that the outcome of DM in RA patients is stated: however, I miss data on radiological damage, orthopedic operations, CV-events etc.
Patient with a cardiovascular event before were excluded, that is in my opinion a flaw: probably the patients with the most severe effects on RA is excluded, why not enrolling all of these patients?
Other points: the longterm observational studies from Prof MT Nurmohamed are missing, on cardiovascular risk in RA and DM.
the study is performed in academic centres, probably with patients with more severe RA, that could be somewhat different in non-academic hospitals.
it is interesting to perform a study in so many academic centres, can the authors add some details about the initiating phase, who started it up, drivers from inside the academia, sponsores form outside? maybe others can learn from the initiation of this large and important study.
Author Response
Dear Reviewer,
We thank you for your suggestions, below you can find the answer to your comments.
Best Regards,
Fabio Cacciapaglia, MD PhD
REVIEWER #2
Interesting work on a large number of RA-patients from Italian academic centres. Interesting data on higher BMI and lower GC use in RA-patients with DM, as expected.
major comments:
- In the introduction it is stated that the outcome of DM in RA patients is stated: however, I miss data on radiological damage, orthopedic operations, CV-events etc.
Answer:
We are sorry if the aim of the study was not clear. At the end of the Introduction, we rephrase the aim of the study as: “Therefore, the aim of the present study was to analyse the demographic and clinical features of RA patients with and without diabetes, describing possible differences in therapeutic management, eventually responsible for a different profile of CV complications”.
- Patient with a cardiovascular event before were excluded, that is in my opinion a flaw: probably the patients with the most severe effects on RA is excluded, why not enrolling all of these patients?
Answer:
As you highlighted, the exclusion of patients with previous CV events might appear to be a bias. However, as the impact of a previous CV event on subsequent CV risk is well known, we opted for the evaluation of patients without a history of CV events to reduce the potential bias on the clinical management of RA and DM: patients with previous CV events have a peculiar therapeutic strategy and specific follow-up that might influence the clinical management of the underlying disease, both for RA and DM.
We added this consideration in the Discussion section when addressing the possible limitations of the study.
Other points:
- the longterm observational studies from Prof MT Nurmohamed are missing, on cardiovascular risk in RA and DM.
Answer:
We appreciate your suggestion, and as rationale for our study we added in the revised version of the manuscript (in the Introduction section) three longitudinal studies on RA and Diabetes from the group of Prof. MT Nurmohamed (J Rheumatol. 2020 - Ann Rheum Dis. 2009 - Arthritis Rheum. 2009)
- the study is performed in academic centres, probably with patients with more severe RA, that could be somewhat different in non-academic hospitals.
Answer:
We understand the point you raised. The inclusion of consecutive RA patients treated according to local (Italian) indications in tertiary-level Rheumatology centers may have selected a population with more severe disease. However, the evidence that about 40% of the entire study population was treated with bDMARDs represents a balanced proportion of patients that would certainly have been higher if only patients with a more severe disease had been selected. This consideration was also added to the Discussion.
- it is interesting to perform a study in so many academic centres, can the authors add some details about the initiating phase, who started it up, drivers from inside the academia, sponsores form outside? maybe others can learn from the initiation of this large and important study.
Answer:
Thanks to the opportunity given by your comment we have better described the initiative that drove the data collection from the beginning. We added the following sentence in the Methods Section “The CORDIS is a non-profit study group established within the Italian Society of Rheumatology on the initiative of academic rheumatologists interested in the study of CV risk in rheumatic diseases. The aim of the group is to understand the interrelation between inflammation, autoimmunity and CVD by collecting epidemiological, clinical and laboratory data from Italian patients with musculoskeletal rheumatic diseases”.
Round 2
Reviewer 2 Report
no additional comments in second round
Author Response
We thank you and, based on your suggestions, we have performed a spell-check and changed all p-values in Table 1 and also the corresponding p-values in the text of the manuscript.
